# Processing Speed and Attentional Shift/Mental Flexibility in Patients with Stroke: A Comprehensive Review on the Trail Making Test in Stroke Studies

Anna Tsiakiri [1,†], Foteini Christidi [1,†], Dimitrios Tsiptsios [1,*], Pinelopi Vlotinou [1], Sofia Kitmeridou [1], Paschalina Bebeletsi [1], Christos Kokkotis [2], Aspasia Serdari [3], Konstantinos Tsamakis [4], Nikolaos Aggelousis [2] and Konstantinos Vadikolias [1]

1   Neurology Department, School of Medicine, Democritus University of Thrace, 681 00 Alexandroupolis, Greece; atsiakir@med.duth.gr (A.T.); christidi.f.a@gmail.com (F.C.); pakivlot@yahoo.com (P.V.); s.kitmer@gmail.com (S.K.); bebeletsi@gmail.com (P.B.); vadikosm@yahoo.com (K.V.)
2   Department of Physical Education and Sport Science, Democritus University of Thrace, 691 00 Komotini, Greece; chkokkotis@gmail.com (C.K.); nagelous@phyed.duth.gr (N.A.)
3   Department of Child and Adolescent Psychiatry, School of Medicine, Democritus University of Thrace, 681 00 Alexandroupolis, Greece; aserntar@med.duth.gr
4   Institute of Psychiatry, Psychology and Neuroscience (IoPPN), King's College London, London SE5 8AB, UK; ktsamakis@gmail.com
*   Correspondence: tsiptsios.dimitrios@yahoo.gr
†   These authors contributed equally to this work.

**Abstract:** The Trail Making Test (TMT) is one of the most commonly administered tests in clinical and research neuropsychological settings. The two parts of the test (part A (TMT-A) and part B (TMT-B)) enable the evaluation of visuoperceptual tracking and processing speed (TMT-A), as well as divided attention, set-shifting and cognitive flexibility (TMT-B). The main cognitive processes that are assessed using TMT, i.e., processing speed, divided attention, and cognitive flexibility, are often affected in patients with stroke. Considering the wide use of TMT in research and clinical settings since its introduction in neuropsychological practice, the purpose of our review was to provide a comprehensive overview of the use of TMT in stroke patients. We present the most representative studies assessing processing speed and attentional shift/mental flexibility in stroke settings using TMT and applying scoring methods relying on conventional TMT scores (e.g., time-to-complete part A and part B), as well as derived measures (e.g., TMT-(B-A) difference score, TMT-(B/A) ratio score, errors in part A and part B). We summarize the cognitive processes commonly associated with TMT performance in stroke patients (e.g., executive functions), lesion characteristics and neuroanatomical underpinning of TMT performance post-stroke, the association between TMT performance and patients' instrumental activities of daily living, motor difficulties, speech difficulties, and mood statue, as well as their driving ability. We also highlight how TMT can serve as an objective marker of post-stroke cognitive recovery following the implementation of interventions. Our comprehensive review underscores that the TMT stands as an invaluable asset in the stroke assessment toolkit, contributing nuanced insights into diverse cognitive, functional, and emotional dimensions. As research progresses, continued exploration of the TMT potential across these domains is encouraged, fostering a deeper comprehension of post-stroke dynamics and enhancing patient-centered care across hospitals, rehabilitation centers, research institutions, and community health settings. Its integration into both research and clinical practice reaffirms TMT status as an indispensable instrument in stroke-related evaluations, enabling holistic insights that extend beyond traditional neurological assessments.

**Keywords:** stroke; Trail Making Test; processing speed; divided attention; cognitive flexibility

## 1. Introduction

The Trail Making Test (TMT) is one of the most commonly administered tests in clinical and research neuropsychological settings. The two parts of the test (part A (TMT-A) and part B (TMT-B)) enable the evaluation of visuoperceptual tracking and processing speed (TMT-A), as well as divided attention, set-shifting, and cognitive flexibility (TMT-B) [1]. Time to complete TMT-A and TMT-B is the most commonly used measure of TMT [1,2]; slowed TMT-B performance compared to TMT-A performance can be a sign of impaired capacity to modify a plan of action and simultaneously keep two streams of thought [3]. In addition to these direct TMT measures, derived scores have become popular in the last decades as sensitive measures of cognitive flexibility and executive dysfunction [4–6]. The most popular derived TMT measures are the difference score (TMT-B − TMT-A; TMT-(B-A)), the ratio score (TMT-B/TMT-A; TMT-(B/A)), and the proportional score (TMT-B − TMT-A)/TMT-A; TMT-(B-A/A). Specifically, it has been suggested that the TMT-(B-A) difference score removes the speed component, minimizes the visuoperceptual and motor demands, and better assesses the executive control processes [7], while the TMT-(B/A) ratio score diminishes the influence of psychomotor demands and controls for factors related to intrasubject variability [5]. Apart from these derived measures, the examination of different error rates in TMT-A and mostly in TMT-B, i.e., sequential and perseverative errors, yield valuable information regarding the cognitive mechanisms of processing speed, attention, and cognitive flexibility [2]. Even though a variety of brain pathologies may influence TMT performance [1,2], TMT is widely used to evaluate executive dysfunction and prefrontal pathology [8], while recent neuroimaging studies highlight the importance of other brain areas as well, such as the parietal cortex [9]. Of note, the executive network includes the frontal lobe, the lateral parietal lobe, subcortical structures (e.g., anterior thalamus, caudate nucleus), and the cerebellum [10–13]. All these areas are connected through white matter tracts, and thus, damage to these structures or their white matter connections could result in slow processing speed and executive dysfunction [14–16].

Cognitive impairment is a common finding among patients with stroke. The prevalence of cognitive impairment is estimated to be between two-thirds and three-quarters of stroke survivors, depending on the methods applied to assess cognition as well as the timing of cognitive assessment, i.e., acute, sub-acute, or chronic phase [17–19]. Post-stroke cognitive impairment is associated with early and enduring changes in patients' daily living activities and quality of life, and this association is still present despite methodological differences in the study design factors (e.g., sample size, patient age, follow-up period) [20]. The main cognitive processes that are assessed using TMT, i.e., processing speed, divided attention, and cognitive flexibility (i.e., one of the executive processes), are often affected in patients with stroke. Of note, patients with stroke due to frontal lesions show greater executive dysfunction than patients with lesions in other regions [21]. Decreased processing speed is an underlying cause of post-stroke cognitive dysfunction [22], while executive dysfunction occurs in the majority of stroke patients [23], posing critical obstacles to the quality of life of these patients [20]. Processing speed and attention deficits remain in the chronic phase [24], whereas the presence of executive dysfunction early after the stroke significantly predicts a poor functional outcome one year post-stroke [25]. Previous studies have also demonstrated an association between executive dysfunction and motor function in stroke [26], including walking and balance ability, postural control, and gait [27–30]. On the other hand, processing speed and attention are associated with on-road driving performance [31] and driving simulator data [32], while executive dysfunction has a significant impact on driving performance since it diminishes individual resources and risk awareness [33]. Such an impact has also been reported in stroke survivors [34].

Considering the wide use of TMT in research and clinical settings since its introduction in neuropsychological practice, the purpose of our review was to provide a comprehensive review of the use of TMT in stroke patients in an attempt to mainly identify its role for the evaluation of post-stroke cognitive dysfunction and progression over time, the identification of the underlying neuroanatomical pathology related to impaired TMT performance, and

the association with other stroke outcomes, such motor function, driving ability and quality of life.

## 2. TMT in Stroke

### 2.1. TMT: Social and Demographic Factors Associated with the Increased Risk of Executive Dysfunction after Stroke

The TMT has been used in a variety of research protocols as part of a broader study design of executive testing due to the fact that post-stroke executive dysfunction is a significant and independent predictor of functional outcome [35–37]. Demographic factors, such as gender and educational level, have been correlated with the increased risk of executive dysfunction after stroke [38,39], while greater cognitive reserve through leisure activity engagement across life can mitigate the negative effects of stroke [40]. Planning [41] and goal-setting [42] after stroke are complex processes and require a thorough examination with a series of assessments to give a more complete picture, taking into account the structural damage caused by the disease. Along the same lines, Cardoso and colleagues emphasize the multidimensionality of executive functions and propose a degree of independence between logic-based and emotion-based executive processes, which should be more thoroughly investigated [43]. Another finding that has not been thoroughly studied is the significant negative effect of binge drinking on the performance of TMT [44].

Addressing the use of the TMT among much younger stroke patients with comorbidities presents several nuanced considerations [45]. Younger individuals experiencing stroke may have distinct comorbidities and varying stroke etiologies, influencing cognitive function differently [46]. Additionally, the greater neuroplasticity in younger brains may impact recovery potential, complicating the interpretation of TMT scores [47]. The role of cognitive reserve and lifestyle factors becomes crucial as younger patients may possess different coping mechanisms [48]. Psychosocial and vocational implications need to be considered, recognizing the unique challenges faced by younger survivors [49].

### 2.2. TMT and Clinical Features

The TMT stands as a cornerstone in neuropsychological assessment, providing valuable insights into cognitive function, particularly executive functions and attention. Recent studies have delved into the intricate interplay between TMT performance and an array of clinical features, unveiling a diverse landscape of connections in various patient populations.

In a study by Einstad and colleagues, completion time exceeding 167 s on TMT-B emerged as a key indicator of impaired executive function, shedding light on a threshold for executive dysfunction in a significant proportion of patients [26]. Mobility and grip strength were intriguingly linked to global cognitive impairments, executive dysfunction, and memory impairment, whereas a higher dual-task cost was specifically correlated with executive dysfunction assessed by TMT-B. Bian and colleagues brought cerebrovascular health into focus, showcasing how the breath-holding index could serve as an insightful parameter for evaluating cerebrovascular reserve impairment, particularly in individuals with leukoaraiosis [50]. This finding adds depth to our understanding of the cognitive consequences of cerebrovascular health. Jo and colleagues added a new layer of complexity by demonstrating the interdependence of cognitive function and post-stroke dysphagia severity [51]. Visual attention and executive functions were pinpointed as influential factors in the oral phase of swallowing, emphasizing the intricate connection between cognitive and motor processes.

The predictive power of cognitive assessment tools took center stage in another study where brief screening instruments exhibited consistent utility, unlike more domain-specific cognitive tests [52]. This highlights the importance of selecting appropriate tools for accurate cognitive assessment in different clinical contexts. Sörös and colleagues underscored the limitations of the Mini-Mental State Examination (MMSE) as an independent cognitive screening tool for transient ischemic attack (TIA) and minor stroke patients [53]. The preva-

lence of executive dysfunction, particularly in TMT Part A and Part B, further emphasized the need for comprehensive cognitive assessment tools. Pedersen and colleagues connected fibrinogen concentrations to cognitive outcomes, emphasizing the role of this factor in younger stroke patients' cognitive performance [54]. Interestingly, the association was specific to TMT-A, indicating distinct patterns of cognitive impairment related to different cognitive tasks.

In the realm of treatment outcomes, Lattanzi and colleagues demonstrated the potential benefits of endovascular treatment combined with intravenous thrombolysis on cognitive performance, hinting at promising avenues for enhancing cognitive outcomes in ischemic stroke patients [55]. Different perspectives opened doors to understanding several factors driving TMT performance post-stroke. Two recent studies underscored the impact of genetic and clinical factors on cognitive outcomes [56,57]. Genetic polymorphisms and clinical conditions like atrial fibrillation were tied to executive dysfunction and cognitive performance, providing a holistic view of the factors shaping cognitive outcomes. The links between blood pressure and cognitive performance were also examined, revealing a more refined relationship independent of sociodemographic and clinical factors, further contributing to our understanding of the complex relationship between blood pressure and cognitive health [58].

According to potential biomarkers, Shaheen and colleagues explored the correlation between serum levels of IL-8 and executive functions in early acute ischemic stroke patients, opening new avenues for understanding the biological underpinnings of cognitive impairment [59]. Rosenbaum Halevi and colleagues hinted at the potential impact of treatment strategies, suggesting a potential cognitive improvement in some cognitive tests at 90 days post-stroke, highlighting the dynamic nature of cognitive recovery [60]. The intricate relationship between cerebral blood flow and cognitive function was demonstrated by Altmann and colleagues, indicating that transcranial Doppler ultrasonography could serve as an early diagnostic tool for cognitive impairment post-stroke [61]. Another study ventured into pharmacological interventions, showcasing how fluoxetine administration might influence cognitive function and serum levels of neurotrophic factors in patients with vascular cognitive impairment [62]. The longitudinal perspective was provided by Ling and colleagues, who highlighted the predictive power of systolic blood pressure and lacunes in assessing cognitive outcomes over time in a specific patient population [63]. Finally, Kotlega and colleagues delved into fatty acid metabolism's role in cognitive outcomes, underscoring the multifaceted interplay between metabolic cascades and stroke-related cognitive impairment [64].

The collective findings of these studies underscore the significance of TMT in unraveling the intricate relationships between cognitive function and various clinical factors in the stroke population. The TMT serves not only as an assessment tool but as a window into the intricate web of cognitive health, offering insights that pave the way for tailored interventions and a deeper understanding of cognitive outcomes post-stroke.

### 2.3. TMT and Neuroanatomical Features

The interplay between cognitive functions and neuroanatomical features has become a focal point for researchers seeking to comprehend the underlying mechanisms of cognitive impairments resulting from brain lesions.

An intriguing trajectory of TMT-A performance improvement from acute to subacute phases following stroke is suggested by Dacosta-Aguayo and colleagues [65]. Notably, better TMT-A performance aligns with higher fractional anisotropy (FA) in specific white matter tracts, offering insights into the relationship between white matter integrity and cognitive recovery. Another study unveiled the association between TMT-A completion time and medial temporal lobe atrophy, as well as global cortical atrophy and lower education levels [66]. These findings highlight the potential of structural brain changes and educational background as rapid indicators of cognitive impairment following transient ischemic attacks and mild ischemic strokes. The study by Cipolotti and colleagues challenges

conventional categorizations by demonstrating no significant differences in frontal executive tasks among different etiology subgroups [67]. Instead, strong effects of premorbid IQ and age on cognitive tasks suggest the influence of broader cognitive factors on executive function. The intricate relationship between executive function, white matter integrity, stroke characteristics, and cerebrovascular risk is investigated by Veldsman and colleagues, who emphasize the mediating role of white matter integrity, highlighting its significance in explaining executive dysfunction and incident stroke, which are both manifestations of cerebrovascular risk factors [68]. Hagberg and colleagues focus on the relationship between TMT-A and amyloid deposition and identify that cortical amyloid deposition does not significantly correlate with neurodegeneration or cognition in stroke survivors with cognitive impairment, emphasizing the complexity of post-stroke cognitive decline and the need to explore additional factors beyond amyloid pathology [69]. A more recent study highlights the potential for brain compensatory mechanisms following stroke [70]. Increased degree centrality values in the right parahippocampal gyrus correlate positively with TMT-A and TMT-B scores, suggesting the brain's adaptive capacity to promote cognitive recovery.

The neural underpinnings of TMT performance are explored by Singh and colleagues, highlighting the intricate connection between spatial planning, working memory, and visual search processes [71]. A recent longitudinal exploration adds a temporal dimension, demonstrating how incident lacunes are independently associated with incident stroke and changes in TMT Part B performance. This longitudinal perspective underscores the long-lasting consequences of structural changes and their impact on cognitive trajectories. Varjačić and colleagues focus on executive set-switching and underscore the critical role of the left insular cortex [72]. By identifying the association between lesions in this region and poorer executive set-switching, the study provides valuable insights into the neural underpinnings of attentional flexibility. Furthermore, the persistence of the lesion effect, even after accounting for lower-level cognitive processes, suggests the insula involvement in higher-order regulatory functions. Another research delves into mental flexibility deficits and their relationship with damaged neural connections [73]. The study unveils the intricate connectivity patterns linking various cortical and subcortical structures implicated in cognitive control and attention networks. An investigation into bimanual grasp coordination following hemispheric strokes underscores the intricate relationship between cognitive and motor functions [74]. The study describes how lesion side and cognitive processes interact to influence motor coordination, providing insights into the interplay between perception, action planning, and lesion site. Jankowska and colleagues' study highlights the wide-ranging effects of stroke locations on executive dysfunction. Contrary to expectations, executive dysfunction is not confined to anterior stroke locations; even posterior and subcortical lesions contribute to the impairment [75]. This underlines the necessity for tailored treatments based on lesion locations. Shin and colleagues examine cerebellar strokes and reinforce the role of cerebellar sites in neuropsychological functioning [76]. The presence of lesions in the right posterior intermediate lobe of the cerebellum correlates with poorer performance in subtests evaluating executive function, such as TMT, shedding light on the specific cognitive consequences of cerebellar lesions. The identification of correlations with the TMT performance underscores the importance of considering connectomics in lesion-symptom mapping, reinforcing the interdependence of regional structures in shaping cognitive outcomes.

In a recent study, Ferris and colleagues introduce DTI metrics of the anterior thalamic radiation as potential imaging biomarkers of post-stroke cognitive impairment [77]. The association between ATR microstructure and processing speed and executive function performance underscores the value of lesion location-specific metrics in predicting cognitive outcomes. Muir's findings reveal associations between larger infarcts, CHIPS severity, and various metrics of set-shifting and processing speed. Moreover, the association between left superior longitudinal fasciculus damage and TMT-(B-A) score further elucidates the role of specific white matter tracts in cognitive performance [78]. Another study utilizes cluster analysis to classify TMT-B performance groups based on derived measures and

identifies brain sites associated with different performance levels, emphasizing the role of specific neural structures in shaping cognitive outcomes during TMT-B completion [79]. In addition, Kopp and colleagues shed light on the patterns of errors in the TMT-A and TMT-B tasks [80] in association with lesion patterns. They identify that the number of errors, but not completion time on the TMT-B, is associated with right hemispheric frontal lesions, while the prevalence of type B shifting errors suggests a failure to switch between numbers and letters, indicating the complex nature of cognitive processes involved in these tasks.

In conclusion, the diverse array of studies examining the relationship between the TMT and neuroanatomical features has provided a multifaceted perspective on the intricate interplay between brain lesions and cognitive functions. These investigations have underscored the localized nature of cognitive deficits, revealed the significance of specific brain regions such as the insular cortex and parahippocampal gyrus, and illuminated the dynamic role of white matter integrity in shaping post-stroke cognitive outcomes. Additionally, the varying impact of lesion side, cognitive domains, and anatomical sites has challenged conventional categorizations and emphasized the need for tailored interventions. As our understanding of the complex relationships between brain structure and cognitive function continues to deepen, these findings offer promising avenues for refining rehabilitation strategies, personalized treatment plans, and imaging biomarkers to mitigate the cognitive consequences of brain lesions.

### 2.4. TMT and Speech Abilities

A recent study highlighted the usefulness of the TMT processing speed component in assessing the underlying processes related to verbal fluency [81]. Of interest, the association between TMT processing speed component and verbal fluency remained significant even after controlling for motor deficits and dysarthria in stroke patients. The study identified shared cognitive processes contributing to fluency tasks. Lesion analysis highlighted the role of left lesions involving deep hemispheric structures and specific brain areas in verbal fluency tasks. In the study of Rajtar-Zembaty and colleagues, patients with aphasia demonstrated a notably higher number of errors in TMT-B compared to non-aphasia and dysarthria groups. This deficit in cognitive flexibility within the aphasia group was associated with dysfunction in the prefrontal cortex, which plays a role in both language skills and components of executive functions [82]. Another study indicated that TMT performance, particularly in TMT-B, was below normal for all groups, with only a subset of aphasics successfully completing the test. The study suggested that TMT might not be the most suitable tool for evaluating left hemisphere (LH) damage and aphasic patients due to their specific impairments [83]. Moreover, Niessen and colleagues suggested that despite clinically relevant cognitive deficits, including aphasia and apraxia, behavioral impairments related to performance monitoring and error processing in LH stroke patients, as measured by the TMT, were not evident; executive dysfunction was present based on TMT scores, but this did not directly translate into observed behavioral impairments in performance monitoring and error processing [84].

The aforementioned studies collectively suggest that the relationship between the TMT and speech abilities, particularly in aphasic patients, involves cognitive flexibility, prefrontal cortex dysfunction, and common neuroanatomical circuits associated with language skills and executive functions. The suitability of the TMT as an evaluation tool for aphasic patients varies, and despite observed executive dysfunction, it may not always translate to significant behavioral impairments in certain tasks.

### 2.5. TMT and Mood Status

The relationship between the TMT and mood status has been explored through various studies, revealing insights into cognitive function and emotional well-being. Donnellan and colleagues did not find any significant correlation between the TMT-(B-A) difference score and the Hospital Anxiety and Depression Scale (HADS) [85]. The absence of correlations suggests that the TMT difference score might not be directly linked to mood disturbances

as assessed by the HADS. Another study examined the relationship between diabetes, depressive symptoms, stroke severity, and TMT performance [86]. The comorbidity index of diabetes and depressive symptoms was associated with poorer performance on the TMT-B (cognitive flexibility and task-switching abilities). However, no significant association was observed between the comorbidity index and TMT-A (visual attention and processing speed). Stroke severity and time since stroke were additional predictors of both TMT-A and TMT-B scores, suggesting complex interactions between medical and mood-related factors in determining cognitive performance. Patients with post-stroke apathy experienced more pronounced cognitive impairments and deficits in attention and executive functions; apathy scale scores were correlated with TMT-A scores at admission and both TMT-A and TMT-B scores, along with other cognitive tests at discharge [87]. Overall, these studies contribute to a deeper understanding of how TMT performance is influenced by and may influence mood-related factors in various contexts.

### 2.6. TMT and Driving Ability

Neuropsychological test performance on tests that measure cognitive/psychomotor speed (TMT A) [88] and executive functioning (TMT B) [34] are the most suitable metrics for predicting driving test outcomes. Some researchers provide optimal cut-off points for TMT-A and TMT-B that can predict post-stroke unsafe driving, i.e., 32 s for TMT-A and 79 s for TMT-B [89]. The combination of TMT with other neuropsychological tests, such as the Snellgrove Maze Task [90], the Useful Field of View test for lane maintenance [91], the Symbol Digit Modalities Test [92], the Rapid Pace Walk test [93] and the Key Search Test of the BADS [34], provides additional data for the prediction of driving ability. Another variant of the TMT is the version in the Delis-Kaplan Executive Function System (D-KEFS; [94]). It has additional tasks on visual search, processing speed, and motor speed, requiring higher levels of visual exploration and is considered to be a significant predictor of offline motor learning [95]. Another reliable variant of TMT-B presented by Lee and colleagues, i.e., driving TMT-B (DTMT-B), used a driving simulator in three-dimensional spaces to test the executive functions of drivers [96].

### 2.7. TMT and Instrumental Activities of Daily Living

The utilization of the TMT-B has proven valuable in evaluating functional autonomy, extending its applicability beyond the post-acute phase. Notably, it has demonstrated a modest yet significant ability to anticipate potential constraints in cognitively rooted Instrumental Activities of Daily Living (IADLs), thus playing a pivotal role in shaping post-discharge treatment strategies, as demonstrated by Minor and colleagues [97]. Similarly, Lipskaya-Velikovsky and colleagues embarked on a parallel investigation to assess intricate routine activities that encompassed tasks demanding advanced planning and vigilant monitoring [36]. The direct TMT measures (time-to-complete TMT-A and TMT-B) did not consistently yield dependable prognosticators of functional proficiency. Ghaffari and colleagues, however, brought to light a noteworthy alternative by highlighting the TMT (B-A) difference score as the singularly reliable predictor for achieving autonomy in IADL performance [98]. The collective insights from these studies illuminate the multifaceted utility of TMT-B, not only in delineating functional autonomy and limitations but also in guiding therapeutic pathways and interventions tailored to enhance cognitive and practical independence.

### 2.8. TMT and Gait Assessment

Numerous studies highlight the interplay between attention, processing speed, and cognitive flexibility, as assessed by the TMT and gait assessment in stroke patients. Executive dysfunction, a common post-stroke sequel, is often linked to compromised gait performance [27,99]. Poorer performance on complex gait tests is frequently associated with worse scores on the TMT, reflecting a shared vulnerability in cognitive and motor domains [100,101]. Dual-task training interventions targeting both cognitive tasks and

gait tasks have consistently demonstrated positive effects on executive function and balance, reinforcing the connection between cognitive demands and gait control in stroke survivors [102,103]. Despite the reported significant associations between TMT scores and gait performance, which highlight the importance of processing speed and executive functions in walking abilities [104,105], there are negative findings as well supporting that gait performance is not universally linked to cognitive function, as evidenced by studies reporting no significant correlation between community ambulation and executive function [106]. This divergence underscores the complexity of their interaction, hinting at potential modulating factors that determine the extent of post-stroke cognitive-motor interdependence [107–109].

### 2.9. TMT in Interventional Studies

The body of research encompassing interventional studies employing the TMT to assess cognitive function in stroke patients presents a compelling narrative of diverse approaches yielding positive outcomes [110–122]. Collectively, these studies underscore the potential of targeted interventions to enhance cognitive abilities, with the TMT serving as a robust indicator of progress. Another study emphasizes the significance of comprehensive training interventions, revealing significant improvements in various measures, including TMT performance, grip strength, and motor function [123]. Another investigation into virtual reality interventions highlights the role of sensory-motor stimulation in fostering improvements in TMT-A and TMT-B scores [124]. Moreover, Gjellesvik and colleagues explored the enduring benefits of interventions by reporting sustained improvements in TMT-B completion time, potentially attributed to heightened arousal and cerebral blood volume changes [125]. These findings are buttressed by the results from a recent study, underscoring the efficacy of cognitive rehabilitation in enhancing cognitive measures, including TMT-A and TMT-B [126]. The significance of intervention adherence surfaces in the Ihle-Hansen study, offering an essential aspect to consider in designing effective cognitive enhancement programs [40]. Interestingly, the advent of technology, such as virtual reality and interactive video games, introduces a novel dimension to cognitive rehabilitation [127]. Such interventions not only foster improvements in TMT performance but also underscore the potential for technology-driven approaches to positively impact executive function. While the majority of studies point to positive outcomes, a few studies highlight the complexities of intervention effects [128]. The potential ceiling effect observed in control groups raises questions about the nuances of experimental design and participant characteristics that warrant further investigation.

The TMT emerges as a pivotal tool within interventional studies targeting cognitive function in stroke patients. Its consistent inclusion across diverse interventions underscores its significance as a sensitive measure for assessing cognitive improvements. The TMT's ability to capture changes in processing speed, attention, and executive function provides researchers with a standardized and quantifiable metric to gauge the efficacy of interventions. Its wide applicability, as demonstrated in studies ranging from cognitive rehabilitation to technology-driven approaches, highlights the test's versatility in evaluating various intervention modalities. Consequently, the TMT not only serves as an objective marker of progress but also enables comparisons across studies, facilitating a comprehensive understanding of the nuanced impacts of interventions on cognitive recovery post-stroke.

### 2.10. Limitations of TMT Performance in Stroke Settings

It is important to acknowledge certain limitations associated with TMT performance. Firstly, TMT performance can be influenced by factors such as age, education, and cultural background, which may impact the generalizability of findings across diverse populations. Moreover, TMT primarily assesses executive functions and attention, potentially overlooking other cognitive domains relevant to stroke outcomes, such as memory or social cognition. Additionally, TMT sensitivity to specific cognitive deficits may vary, and its

ability to capture subtle changes in cognitive function over time might be limited. The interpretation of TMT scores also requires careful consideration of individual differences in baseline abilities. Furthermore, TMT's reliance on paper and pencil may not fully reflect real-world scenarios, limiting its ecological validity. Despite its widespread use, it is essential to recognize these limitations when interpreting results and consider complementary assessments to obtain a more comprehensive understanding of post-stroke cognitive function.

### 3. Discussion

The present literature review attempts to describe and analyze the use and application of the TMT in the clinical field of stroke. To the best of our knowledge, no such attempt has previously occurred despite the widespread use of TMT in stroke studies. Through a comprehensive review of the literature in the last decade, we provide a conceptual framework of the available findings, further highlighting potential gaps and challenges of the application of TMT in clinical and research settings (Figure 1).

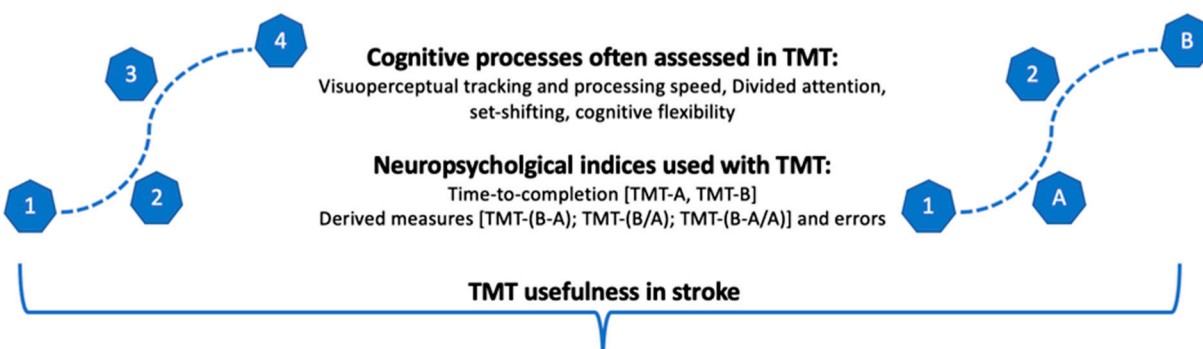

**Figure 1.** A graphical representation of the cognitive processes assessed using TMT and the main neuropsychological indices of the test, as well as its usefulness in stroke studies.

The association between TMT performance and IADLs underscores the clinical relevance of TMT in predicting individuals' capacity to navigate daily life tasks independently, making it pertinent for rehabilitation centers and long-term care facilities. Moreover, its role in assessing driving ability reveals its potential for safeguarding not only the patient but also the broader community, indicating its importance in assessing fitness to drive and informing licensing authorities.

The integration of TMT into executive function testing illuminates its pivotal role in revealing deficits in higher-order cognitive processes, aiding clinicians in formulating tailored rehabilitation strategies for stroke survivors seeking to regain cognitive and functional independence. Its linkage to gait assessment emphasizes the intricate interplay between cognitive and motor functions, offering novel insights into post-stroke mobility issues, thereby finding application in physical therapy and geriatric care settings. Of note, the effectiveness of new technologies, such as exoskeleton-assisted gait [129] and robotic-assisted rehabilitation [130], may be highly related to adequate patients' mental processing speed, attentional shift, and cognitive flexibility status, and thus such knowledge may assist in better patients' stratification and goal setting identification both at baseline and follow-up

The correlation between TMT outcomes and speech abilities underscores its sensitivity to language-related impairments post-stroke, making it a crucial tool in speech therapy in-

terventions. Equally noteworthy is the potential of TMT to capture mood status, facilitating a more comprehensive understanding of the emotional toll of stroke and aiding in holistic patient management, which is relevant in both clinical and psychological support contexts.

The incorporation of TMT within interventional studies holds promise for refining therapeutic approaches and measuring treatment efficacy, contributing to the advancement of stroke rehabilitation protocols in research institutions and clinical trials. Its robustness in capturing clinical and neuroanatomical features further underscores its potential as a biomarker for stroke severity and lesion localization, which has implications for both academic research and diagnostic settings. Considering the role of endogenic factors in stroke prevalence, such as gene polymorphisms [131], the synergistic effect of specific genotypes on processing speed and attentional shift/mental flexibility post-stroke should be examined.

Therefore, our comprehensive review underscores that the TMT stands as an invaluable asset in the stroke assessment toolkit, contributing nuanced insights into diverse cognitive, functional, and emotional dimensions. As research progresses, continued exploration of the TMT potential across these domains is encouraged, fostering a deeper comprehension of post-stroke dynamics and enhancing patient-centered care across hospitals, rehabilitation centers, research institutions, and community health settings. Its integration into both research and clinical practice reaffirms TMT status as an indispensable instrument in stroke-related evaluations, enabling holistic insights that extend beyond traditional neurological assessments.

Strengths of this comprehensive review include a rigorous and comprehensive database search covering a wide array of topics, ranging from TMT associations with cognitive and functional outcomes to its implications in various domains, such as driving ability, executive functions, mood status, and more. This breadth of coverage ensures a holistic understanding of TMT's multifaceted utility within the context of stroke assessment. Secondly, the incorporation of diverse sources of evidence, including RCTs, original studies, clinical trials, and theoretical frameworks, lends depth to the analysis and strengthens the conclusions drawn. Additionally, the emphasis on the applicability of TMT findings in clinical and research frameworks underscores the practical relevance of the review's insights. The present review has some limitations that need to be addressed. The review's reliance on existing literature up to a certain knowledge cutoff date could potentially omit relevant studies published afterward. The diversity of stroke populations, varying degrees of stroke severity, and individual patient characteristics might introduce heterogeneity in the results across studies, potentially influencing the generalizability of the findings. The diversity and the number of surveys included did not allow for a quality assessment of the included studies nor a meta-analysis of the results. Due to the comprehensive nature of the present review and the heterogeneity of sample characteristics of different studies, we did not specifically focus on the TMT performance in different stroke types. Future systematic reviews and meta-analyses can certainly capture the full spectrum of cognitive outcomes in processing speed and attentional shift/cognitive flexibility (TMT-part A and TMT-part B, respectively) and enhance the applicability of findings across different stroke etiologies.

Future research endeavors in the realm of the TMT and stroke assessment hold great promise for refining clinical practice and advancing our knowledge. Longitudinal studies tracking TMT performance over time can illuminate cognitive recovery trajectories post-stroke, while investigations into the predictive validity of TMT scores can establish its role as a prognostic tool. Exploring neuroplasticity and cognitive training interventions in relation to TMT performance can uncover avenues for enhancing recovery. Integrating TMT with neuroimaging can deepen our understanding of its neural correlates, and accounting for cultural and linguistic factors can lead to culturally sensitive assessments. The feasibility of technology integration, the role of TMT in special populations, and combining it with other assessments all offer avenues for comprehensive stroke evaluations. Continual standardization efforts and updated norms ensure TMT's consistent and reliable use. Overall, these research directions can enrich stroke rehabilitation strategies, bolster

patient outcomes, and advance our comprehension of cognitive function and post-stroke recovery dynamics.

## 4. Conclusions

In conclusion, the TMT emerges as a versatile and valuable neuropsychological tool with significant implications for stroke-related assessments. Our comprehensive review of its applications within the stroke population highlights its multifaceted utility in evaluating various domains of cognitive and functional abilities, spanning both acute and chronic phases of stroke recovery.

**Author Contributions:** A.T. and F.C. reviewed the literature, screened the abstracts of the reference list, and assessed the articles; D.T. and P.V. solved any disagreement regarding the screening or selection process; A.T. and F.C. wrote the first manuscript; S.K., P.B., C.K., A.S., N.A. and K.V. reviewed the draft and provided valuable feedback. The corrected version was discussed collegially. A.T., K.T. and F.C. updated and wrote the final version. All authors have read and agreed to the published version of the manuscript.

**Funding:** This research was funded by the project "Study of the interrelationships between neuroimaging, neurophysiological and biomechanical biomarkers in stroke rehabilitation (NEURO-BIO-MECH in stroke rehab)" (MIS 5047286), which was implemented under the action "Support for Regional Excellence" funded by the operational program "Competitiveness, Entrepreneurship and Innovation" (NSRFm2014-2020) and co-financed by Greece and the European Union (the European Regional Development Fund).

**Data Availability Statement:** Not applicable.

**Conflicts of Interest:** The authors declare no conflicts of interest.

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
