# Peer review of "Processing Speed and Attentional Shift/Mental Flexibility in Patients with Stroke: A Comprehensive Review on the Trail Making Test in Stroke Studies"

_2035-8377, doi:10.3390/neurolint16010014_

Round 1

Reviewer 1 Report

Comments and Suggestions for Authors

The paper titled “Processing speed and attentional shift/mental flexibility in patients with stroke: A comprehensive review on the Trail Making Test in stroke studies” provided a very through, and useful review on TMT research across the world.  

The authors provided nine aspects on TMT and stroke in this review article. I am not sure the rational with the order of these nine aspects, however, I would feel it would be clearer if the topics followed through the stages after having stroke.

For example, the aspects flow suggested as the following:

(1)   The social and demographic factors associated with the increased risk of executive dysfunction after stroke.

Additional comments: would you address the challenges or insights on TMT among much younger stroke patients with the comorbidities?

(2)   The clinical features, especially the treatment associated with the cognitive outcomes in stroke patients. The authors provided several studies on ischemic stroke patients. It would be more important to address TMT and hemorrhagic stroke patients as well since the etiology is different between ischemic and hemorrhagic stroke.

(3)   TMT and neuroanatomical features

(4)   TMT and speech abilities

(5)   TMT and mood status

(6)   TMT and driving ability

(7)   TMT and instrumental activities of daily living

(8)   TMT and gait assessment

(9)   TMT in interventional study

Author Response

Reviewer 1

Reviewer: The paper titled “Processing speed and attentional shift/mental flexibility in patients with stroke: A comprehensive review on the Trail Making Test in stroke studies” provided a very through, and useful review on TMT research across the world.  

Response: We thank the reviewer for dedicating time to review this manuscript and the encouraging comments with regards to the review and the importance of the topic for the stroke and TMT research across the world. 

Reviewer: The authors provided nine aspects on TMT and stroke in this review article. I am not sure the rational with the order of these nine aspects, however, I would feel it would be clearer if the topics followed through the stages after having stroke. For example, the aspects flow suggested as the following:

(1)   The social and demographic factors associated with the increased risk of executive dysfunction after stroke.

Additional comments: would you address the challenges or insights on TMT among much younger stroke patients with the comorbidities?

(2)   The clinical features, especially the treatment associated with the cognitive outcomes in stroke patients. The authors provided several studies on ischemic stroke patients. It would be more important to address TMT and hemorrhagic stroke patients as well since the etiology is different between ischemic and hemorrhagic stroke.

(3)   TMT and neuroanatomical features

(4)   TMT and speech abilities

(5)   TMT and mood status

(6)   TMT and driving ability

(7)   TMT and instrumental activities of daily living

(8)   TMT and gait assessment

(9)   TMT in interventional study

Response: We would like to thank you for the valuable comment. We made the suggested modifications. We also added some information about the first aspect (1) about young stroke patients, “Addressing the use of the TMT among much younger stroke patients with comorbidities presents several nuanced considerations (10.1038/nrneurol.2014.72). Younger individuals experiencing stroke may have distinct comorbidities and varying stroke etiologies, influencing cognitive function differently (10.1212/WNL.0000000000001882). Additionally, the greater neuroplasticity in younger brains may impact recovery potential, complicating the interpretation of TMT scores (10.1002/ddrr.64). The role of cognitive reserve and lifestyle factors becomes crucial, as younger patients may possess different coping mechanisms (https://doi.org/10.1016/j.arr.2022.101814). Psychosocial and vocational implications need to be considered, recognizing the unique challenges faced by younger survivors. (10.1177/1747493017743059; 10.1097/JNN.0000000000000523).”.

Regarding the comment in the second aspect (2): Our literature search included both types of cerebral infarctions. Based on our initial review, we identified studies that reported one of the two types of stroke , other studies that did not provide any information regarding the type of stroke (e.g. Hayes et al. 2013; Ghaffari et al. 2020; Choi et al. 2016; Kim et al. 2017, and studies that included mixed samples of ischemic and hemorrhagic patients (e.g. Kim et al. 2013; Lee et al. 2020; Kopp et al. 2015; Durcan et al. 2016; Lee et al. 2016; Ghaffari et al. 2021; Einstad et al. 2021; Donnella et al. 2016; Wolf et al. 2013; Lipskaya-Velikovsky et al. 2018. Due to the comprehensive nature of the present review and the heterogeneity of sample characteristics of different studies, we did not specifically focus on the TMT performance in different stroke types. Future systematic reviews and meta-analyses can certainly capture the full spectrum of cognitive outcomes in processing speed and attentional shift/cognitive flexibility (TMT-part A and TMT-part B, respectively) and enhance the applicability of findings across different stroke etiologies. We now added this comment in the Discussion section, “Due to the comprehensive nature of the present review and the heterogeneity of sample characteristics of different studies, we did not specifically focus on the TMT performance in different stroke types. Future systematic reviews and meta-analyses can certainly capture the full spectrum of cognitive outcomes in processing speed and attentional shift/cognitive flexibility (TMT-part A and TMT-part B, respectively) and enhance the applicability of findings across different stroke etiologies.”. Thank you.

Reviewer 2 Report

Comments and Suggestions for Authors

Authors presented an interesting comprehensive review about processing speed and attentional shift/mental flexibility in patients with stroke on the Trail Making Test in stroke studies. The findings of the review reported that Trail Making Test stands as an invaluable asset in the stroke assessment toolkit, contributing nuanced insights into diverse cognitive, functional, and emotional dimensions.
The review is interesting, I suggest to consider the role of endogenic factors in stroke (10.1007/s00439-012-1224-9) and the importance of rehabilitation and, in particular, the role of new technologies in stroke (10.1016/j.clinph.2020.04.158 and 10.26355/eurrev_202309_33580).

Author Response

Reviewer 2: Authors presented an interesting comprehensive review about processing speed and attentional shift/mental flexibility in patients with stroke on the Trail Making Test in stroke studies. The findings of the review reported that Trail Making Test stands as an invaluable asset in the stroke assessment toolkit, contributing nuanced insights into diverse cognitive, functional, and emotional dimensions.

Response: We thank the reviewer for dedicating time to review this manuscript and the encouraging comments with regards to the review and the importance of the topic for the stroke. 

Reviewer 2: The review is interesting, I suggest to consider the role of endogenic factors in stroke (10.1007/s00439-012-1224-9) and the importance of rehabilitation and, in particular, the role of new technologies in stroke (10.1016/j.clinph.2020.04.158 and 10.26355/eurrev_202309_33580).

Response: We really appreciate reviewer’s comment and suggestions. We now included the following sentences in the Discussion section, “Considering the role of endogenic factors in stroke prevalence, such gene polymorphisms [ref. 10.1007/s00439-012-1224-9], the synergistic effect of specific genotypes on processing speed and attentional shift/mental flexibility post-stroke should be examined.and Of note, the effectiveness of new technologies, such as exoskeleton-assisted gait [ref. 10.1016/j.clinph.2020.04.158] and robotic-assisted rehabilitation [10.26355/eurrev_202309_33580], may be highly related to adequate patients’ mental processing speed, attentional shift, and cognitive flexibility status, and thus such a knowledge may assist in better patients’ stratification and goal setting identification both at baseline and follow-up.”

Thank you.

Reviewer 3 Report

Comments and Suggestions for Authors

This paper is a narrative overview of the trail-making test (TMT) and stroke. It describes its association with cognitive abilities, instrumental activities of daily living, driving, gait, speech, mood, role in intervention studies, clinical features, and neuroanatomy. The paper is well-written and summarises the key information on TMT and stroke.

There are 3 issues the authors may wish to address:

1.       Abstract – please add more information on what TMT is associated with wrt cognitive abilities, instrumental activities of daily living, driving, gait, speech, mood, role in intervention studies, clinical features, neuroanatomy – paraphrasing and building on the information in Fig 1 may be helpful

2.       Main text – have a separate paragraph that discusses the limitations of TMT as wat is mentioned is lost within the main body

3.       If allowed, the TMT can be added as a figure

Author Response

Reviewer 3

Reviewer: This paper is a narrative overview of the trail-making test (TMT) and stroke. It describes its association with cognitive abilities, instrumental activities of daily living, driving, gait, speech, mood, role in intervention studies, clinical features, and neuroanatomy. The paper is well-written and summarises the key information on TMT and stroke. There are 3 issues the authors may wish to address:

Response: We thank the reviewer for dedicating time to review this manuscript and the encouraging comments with regards to the review and the importance of the topic for the stroke. We provide a detailed response for each of the three comments raised by the reviewer.

Reviewer:  #1. Abstract – please add more information on what TMT is associated with wrt cognitive abilities, instrumental activities of daily living, driving, gait, speech, mood, role in intervention studies, clinical features, neuroanatomy – paraphrasing and building on the information in Fig 1 may be helpful

Response: We really appreciate reviewer’s comment for the Abstract. Based on the reviewer’s suggestion, we elaborated on the TMT and TMT-associated sections in the abstract. The following sentence has been deleted in an attempt to mainly identify its role for the evaluation of post-stroke cognitive dysfunction and progression over time, the identification of the underlying neuroanatomical pathology related to impaired TMT performance, and the association with other stroke outcomes, such motor function, driving ability and quality of life.” while the following sentences have been included in the Abstract: “We present the most representative studies assessing processing speed and attentional shift/mental flexibility in stroke settings using TMT and applying scoring methods relying on conventional TMT scores (e.g. time-to-complete part A and part B), as well as derived measures (e.g. TMT-(B-A) difference score, TMT-(B/A) ratio score, errors in part A and part B). We summarize the cognitive processes commonly associated with TMT performance in stroke patients (e.g. executive functions), lesion characteristics and neuroanatomical underpinning of TMT performance post-stroke, the association between TMT performance and patients’ instrumental activities of daily living, motor difficulties, speech difficulties, and mood statue, as well as their driving ability. We also highlight how TMT can serve as an objective marker of post-stroke cognitive recovery following the implementation of interventions”. Thank you.

Reviewer 3:  #2.  Main text – have a separate paragraph that discusses the limitations of TMT as wat is mentioned is lost within the main body

Response: We appreciate reviewer’s comment. In the revised version we added a separate section regarding limitations, as suggested by the reviewer. The following paragraph is now included, “2.10. Limitations of TMT performance in stroke settings: It is important to acknowledge certain limitations associated with TMT performance. Firstly, TMT performance can be influenced by factors such as age, education, and cultural background, which may impact the generalizability of findings across diverse populations. Moreover, TMT primarily assesses executive functions and attention, potentially overlooking other cognitive domains relevant to stroke outcomes, such memory or social cognition. Additionally, TMT sensitivity to specific cognitive deficits may vary, and its ability to capture subtle changes in cognitive function over time might be limited. The interpretation of TMT scores also requires careful consideration of individual differences in baseline abilities. Furthermore, TMT reliance on paper and pencil may not fully reflect real-world scenarios, limiting its ecological validity. Despite its widespread use, it is essential to recognize these limitations when interpreting results and consider complementary assessments to obtain a more comprehensive understanding of post-stroke cognitive function.Thank you.

Reviewer 3:  # 3.  If allowed, the TMT can be added as a figure

Response: We do agree that a figure depicting TMT part A and part B material would be useful for the readers. However, as we do not have permission for the reproduction of the image with the original version of the test, it is not included neither in the original nor in the revised version. To roughly showcase the material of the TMT, we used an abstract graphical representation of part A and part B in the original version of the paper, Figure 1.